# Limiting Extrapolation in
# Linear Approximate Value Iteration

**Andrea Zanette**
Institute for Computational and Mathematical Engineering,
Stanford University, CA
zanette@stanford.edu

**Alessandro Lazaric**
Facebook AI Research
lazaric@fb.com

**Mykel J. Kochenderfer**
Department of Aeronautics and Astronautics,
Stanford University, CA
mykel@stanford.edu

**Emma Brunskill**
Department of Computer Science,
Stanford University, CA
ebrun@cs.stanford.edu

## Abstract

We study linear approximate value iteration (LAVI) with a generative model. While linear models may accurately represent the optimal value function using a few parameters, several empirical and theoretical studies show the combination of least-squares projection with the Bellman operator may be expansive, thus leading LAVI to amplify errors over iterations and eventually diverge. We introduce an algorithm that approximates value functions by combining Q-values estimated at a set of *anchor* states. Our algorithm tries to balance the generalization and compactness of linear methods with the small amplification of errors typical of interpolation methods. We prove that if the features at any state can be represented as a convex combination of features at the anchor points, then errors are propagated linearly over iterations (instead of exponentially) and our method achieves a polynomial sample complexity bound in the horizon and the number of anchor points. These findings are confirmed in preliminary simulations in a number of simple problems where a traditional least-square LAVI method diverges.

## 1   Introduction

Impressive empirical successes [Mni+13; Sil+16; Sil+17] in using deep neural networks in reinforcement learning (RL) often use sample inefficient algorithms. Despite recent advances in the theoretical analysis of value-based batch RL with function approximation [MS08; ASM08; FSM10; YXW19; CJ19], designing provably sample-efficient approximate RL algorithms with function approximation remains an open challenge.

In this paper, we study value iteration with linear approximation (LAVI for short). Linear function approximators represent action-value functions as the inner product between a weight vector $w$ and a $d$-dimensional feature map $\phi$ evaluated at each state-action pair, i.e., $\widehat{Q}(s,a) = w^\top \phi(s,a)$. Linear models are common and powerful because they allow to compactly represent functions with a small number of parameters, and therefore have promise for requiring a small sample size to learn such functions. Unfortunately, it is well known that the Bellman operator combined with the projection onto a linear space in, e.g., $\ell_2$-norm, may result in an expansive operator. As a result, even when the features are expressive enough so that the optimal state-action value function $Q^\star$ can be accurately represented (i.e., $Q^\star(s,a) \approx (w^\star)^\top \phi(s,a)$), combining linear function approximation with value iteration may lead to divergence [Bai95; TV96]. Munos [Mun05] derived bounds on the error propagation for general approximate value iteration (AVI) and later Munos and Szepesvári

[MS08] proved finite-sample guarantees for *fitted value iteration* with a generative model, while sharper results can be found in [FSM10]. A key issue in AVI is that errors at one iteration may be amplified through the application of the Bellman operator and projection. In the analysis of Munos and Szepesvári [MS08], this effect is illustrated by the *inherent Bellman error*, which measures how well the image through the Bellman operator of any function in the approximation space can be approximated within the space itself. Whenever the inherent Bellman error is unbounded, AVI may diverge.

In contrast to the amplification of errors of linear value function approximation, averagers [Gor95], such as barycentric interpolators [MM99], nearest-neighbors, and kernels [OS02], can reduce how errors are propagated through iterations. Averagers represent the value function at a state-action pair as an interpolation of its values at a finite set of anchor points. By interpolating instead of extrapolating, the function approximator is guaranteed to be a non-expansion in $\ell_\infty$-norm, and therefore the Bellman backup remains a contraction even after the projection onto the approximation space. Unfortunately, the number of anchor points needed to accurately represent the value function, and thus the number of parameters to learn, may scale exponentially with the input state dimension.

In this paper, we explore a new function approximator that tries to balance the compactness and generalization of linear methods, leading to sample efficiency at each iteration, while constraining the resulting expansion, as in averagers, thus providing a small amplification factor over iterations. Our algorithm estimates the Q-values at a set of anchor points and predict the function at any other point by taking a combination of those values, while using a linear representation. We show that whenever the features generate a convex set, it is possible to avoid any error amplification and achieve a sample complexity that is polynomial in the number of anchor points and in the horizon. A related convexity assumption has been very recently used by Yang and Wang [YW19] to obtain the first algorithm with near-optimal sample complexity. Nonetheless, their result holds when the transition model $p$ admits a non-negative low-rank factorization in $\phi$, which also corresponds to a zero inherent Bellman error. In our analysis, we consider the far more general setting of when the optimal state-action value function can be accurately approximately with a linear set of features. Note that this can be true even if the transition model does not admit a low-rank decomposition, as we illustrate in our simulation results. Furthermore, our result holds even when the inherent Bellman error is infinite. Unlike [YW19], we also report a thorough discussion on how to select anchor points and provide a heuristic procedure to automatically create them.

In our simulations we show that small levels of amplification can be achieved, and that our algorithm can effectively mitigate the divergence observed in some simple MDPs for least-squares AVI. This happens even when using identical feature representations, highlighting the benefit of bounding extrapolation through constructing feature representations as near convex combinations (versus $\ell_2$ or other common loss functions). Furthermore, we empirically show that small amplification factors can be obtained with relatively small sets of anchor points. We believe this work provides a first step towards designing sample efficient algorithms that effectively balance per-iteration generalization and sample complexity and the amplification of errors through iterations for general linear action-value function solvers.

## 2   Preliminaries

We consider a fixed-horizon MDP $\mathcal{M} = \langle \mathcal{S}, \mathcal{A}, p, r, H, \rho \rangle$ defined by a continuous state space $\mathcal{S}$, a discrete action space $\mathcal{A}$, a horizon $H$, an initial state distribution $\rho$, a transition model $p(s, a)$ and a reward model $r(s, a)$. We also denote by $R(s, a)$ the random reward, with expected value $r(s, a)$. A deterministic policy $\pi_t(s)$ is a mapping from a state and timestep to an action. The $Q$-value of a policy $\pi$ in state-action-timestep $(s, a, t)$ is the expected return after taking action $a$ in $s$ at timestep $t$ and following policy $\pi$ afterwards, and $V_t^\pi(s) = Q_t^\pi(s, \pi_t(s))$. An optimal policy $\pi^\star$ maximizes the value function at any state and timestep, i.e., $\pi_t^\star = \arg\max_\pi V_t^\pi$. We use $V_t^\star = V_t^{\pi^\star}$ and $Q_t^\star = Q_t^{\pi^\star}$ to denote the functions corresponding to an optimal policy $\pi^\star$.

We consider the so-called *generative model* setting, where $p$ and $r$ are unknown but a simulator can be queried at any state-action pair $(s, a)$ to obtain samples $s' \sim p(s, a)$ and $R(s, a)$. As the generation of each sample may be expensive, the overall objective is to compute a near-optimal policy with as few samples as possible. Approximate dynamic programming algorithms can be used to replace $p$ and $r$ with a finite number of simulator samples, and can be used for high dimensional or continuous spaces. Approximate value iteration (AVI) (related closely to *fitted value iteration*), takes as input a regression algorithm $\mathcal{F}$, and it proceeds backward from horizon $H$ to 1. At each timestep $t$, given the

approximation $\widehat{Q}^{\star}_{t+1}$, it queries the simulator $n$ times and obtains a set of tuples $\{(s_i, a_i, r_i, s'_i)\}^n_{i=1}$, used to construct a regression dataset $\mathcal{D}_t = \{(s_i, a_i), y_i)\}^n_{i=1}$ with $y_i = r_i + \max_a \widehat{Q}^{\star}_{t+1}(s'_i, a)$. AVI then computes $\widehat{Q}^{\star}_t = \mathcal{F}(\mathcal{D}_t)$, returns the approximated optimal policy $\widehat{\pi}^{\star}_t(s) = \arg\max_a \widehat{Q}^{\star}_t(s, a)$, and proceed to timestep $t - 1$.

A popular instance of AVI is to use linear regression to approximate $Q$-functions. We refer to this general scheme as linear AVI (LAVI). Let $\phi_t : \mathcal{S}_t \times \mathcal{A}_t \to \mathbb{R}^d$ be a feature mapping for timestep $t$. We define $\Phi_t = \{\phi \in \mathbb{R}^d : \exists s \in \mathcal{S}, a \in \mathcal{A}_{t,s}, \phi_t(s, a) = \phi\}$ as the subset of $\mathbb{R}^d$ obtained by evaluating $\phi_t$ at any state-action pair $(s, a)$. Any approximate action-value function $\widehat{Q}_t$ is represented as a linear combination of weights $\widehat{w}_t \in \mathbb{R}^d$ and features $\phi_t$ as $\widehat{Q}_t(s, a) = \widehat{w}^{\top}_t \phi_t(s, a)$, where $\widehat{w}_t$ is usually computed by minimizing the $\ell_2$-loss on the dataset $\mathcal{D}_t$. Linear function approximation requires only $O(d/\epsilon^2)$ samples to have an $\epsilon$ estimation error, independent from the size of $\mathcal{S}$ and $\mathcal{A}$. Nonetheless, at each timestep $t$ the combination of the $\ell_2$-loss minimization (i.e., $\mathcal{F}$) with the application of the Bellman operator to the function computed at timestep $t + 1$ may correspond to an expansive operation. In this case, errors at each iteration may be amplified and eventually lead LAVI to diverge.

# 3 Linear Approximate Value Iteration with Extrapolation Reduction

We introduce IER (*Interpolation for Extrapolation Reduction*), a novel approximation algorithm that interpolates $Q$-values at a set of anchor points. We study its prediction error and we analyze the sample complexity of the LAVI scheme obtained by executing IER backward from $H$ to 1.

At each timestep $t$, IER receives as input an estimate $\widehat{Q}^{\star}_{t+1}$ of the action-value function at timestep $t + 1$, the feature map $\phi_t$, and a set $\mathcal{K}_t \subseteq \mathcal{S} \times \mathcal{A}$ of $K_t$ anchor state-action pairs. IER first estimates $Q^{\star}_t(s_i, a_i)$ at any anchor point $(s_i, a_i) \in \mathcal{K}_t$ by repeatedly sampling from the simulator and using the approximation $\widehat{Q}^{\star}_{t+1}$ to compute the backup values. We define the *anchor values* as

$$\widehat{Q}^{\star}_{t,i} = \frac{1}{n_{\text{supp}}} \sum_{j=1}^{n_{\text{supp}}} \left( R_t^{(j)} + \max_{a \in \mathcal{A}} \widehat{Q}^{\star}_{t+1}(s_{t+1}^{(j)}, a)) \right), \tag{1}$$

where $R_t^{(j)}$ and $s_{t+1}^{(j)}$ are the samples generated from the generative model at $(s_i, a_i)$ and $n_{\text{supp}}$ is the budget at each anchor point. Given these estimations, the approximation $\widehat{Q}^{\star}_t(s, a)$ returned by IER at any state-action pair $(s, a)$ is obtained by a linear combination of the $\widehat{Q}^{\star}_{t,i}$ values as

$$\widehat{Q}^{\star}_t(s, a) = \sum_{i=1}^{K_t} \theta^{\phi_t(s,a)}_{t,i} \widehat{Q}^{\star}_{t,i}, \tag{2}$$

where the interpolation vector $\theta^{\phi_t(s,a)}_t \in \mathbb{R}^K$ is the solution to the optimization problem

$$\min_{\theta^{\phi_t(s,a)}} \|\theta^{\phi_t(s,a)}\|_1 \qquad \text{subject to} \;\; \phi_t(s, a) = \sum_{i=1}^{K_t} \theta^{\phi_t(s,a)}_i \phi_t(s_i, a_i). \tag{3}$$

As long as the image of the anchor points $\{\phi(s_i, a_i)\}^{K_t}_{i=1}$ spans $\mathbb{R}^d$, (3) admits a solution. This problem is a *linear* optimization program with *linear* constraints and it can be solved efficiently using standard techniques [BV04; NW06]. Notice that the weights $\theta^{\phi_t(s,a)}_t$ change with $s, a$ and no positiveness constraint is enforced.

## 3.1 Prediction Error and Sample Complexity of IER

In most problems, the optimal action-value function $Q^{\star}_t$ cannot be exactly represented by a low dimensional inner product $w^{\top}_t \phi_t(\cdot, \cdot)$. The best approximator that can be expressed by features $\phi$ and its associated approximation error are defined as

$$w^{\star}_t = \arg\min_{w \in \mathbb{R}^d} \left\| w^{\top} \phi_t(\cdot) - Q^{\star}_t(\cdot) \right\|_\infty ; \qquad \epsilon^{app}_t = \min_{w \in \mathbb{R}^d} \left\| w^{\top} \phi_t(\cdot) - Q^{\star}_t(\cdot) \right\|_\infty, \tag{4}$$

where $\| \cdot \|_\infty$ denotes the infinity norm, i.e., the maximum over state-action pairs in $\mathcal{S} \times \mathcal{A}$. Standard linear function approximation methods rather minimize the $\ell_2$-norm (i.e., least-squares) or a regularized version of it.

We are interested in studying whether IER approaches the performance of $w^\star$. Before analyzing IER, we focus on its "exact" counterpart. We introduce $\widetilde{Q}_t^\star(s,a)$ as the interpolator obtained by combining the exact $Q^\star$-function evaluated on the anchor points as

$$\widetilde{Q}_t^\star(s,a) \stackrel{def}{=} \sum_{i=1}^{K_t} \theta_i^{\phi(s,a)} Q_t^\star(s_i, a_i) \tag{5}$$

where the vector $\theta^{\phi(s,a)}$ is the solution of (3). We prove the following.

**Lemma 1** (Error Bounds of $\widetilde{Q}_t^\star$). *Let $\epsilon_t^{app}$ be the approximation error of the best linear model (Eq. 4). If $\epsilon_t^{app} = 0$, i.e., $Q_t^\star(s,a) = (w_t^\star)^\top \phi_t(s,a)$, then $\widetilde{Q}_t^\star(s,a) = (w_t^\star)^\top \phi_t(s,a)$. Otherwise the (exact) interpolator in Eq. 5 has an error*

$$\max_{(s,a)\in\mathcal{S}\times\mathcal{A}} \left| \widetilde{Q}_t^\star(s,a) - Q_t^\star(s,a) \right| \le (1 + C_t)\epsilon_t^{app}, \tag{6}$$

*where $C_t \stackrel{def}{=} \max_{(s,a)\in\mathcal{S}\times\mathcal{A}} \|\theta_t^{\phi(s,a)}\|_1$ is the amplification factor.*

This result shows that the interpolation done in (5) preserves the linearity of the model whenever the function evaluated at the anchor points is linear itself. Furthermore, the prediction error is a factor $(1 + C_t)$ bigger than the best approximator. The optimization program (3) plays a crucial role in obtaining both results. In particular, the constraint ensures that the linear structure is preserved, while the minimization over $\theta_t^{\phi(s,a)}$ aims at controlling the amplification factor $C_t$. We now study the sample complexity of IER at timestep $t$ when an approximation of the optimal value function $V_{t+1}^\star$ at timestep $t + 1$ is available (the proof and definition of $\delta'$ is postponed to the supplementary).

**Lemma 2.** *Let $\epsilon_t^{app}$ be the error of the best linear model at timestep $t$ and $\widehat{V}_{t+1}^\star$ be the approximation of $V_{t+1}^\star$ used in estimating the values at the anchor points in Eq. 1. Let $\|\widehat{V}_{t+1}^\star - V_{t+1}^\star\|_\infty \le \epsilon_{t+1}^{bias}$ be the prediction error of $\widehat{V}_{t+1}^\star$. If IER is run with $K_t$ anchor points, then the prediction error of $\widehat{Q}_t^\star$ is*

$$\|\widehat{Q}_t^\star - Q_t^\star\|_\infty \le \underbrace{\left((1 + C_t)\epsilon_t^{app} + C_t\epsilon_t^{est}\right)}_{\text{errors at timestep } t} + \underbrace{C_t\epsilon_{t+1}^{bias}}_{\text{propagation error}} \tag{7}$$

*with probability at least $1 - \delta/H$ as long as $n_{supp} \ge \ln(2/\delta')/(2\epsilon_t^{est})^2$.*

Lem. 2 shows that the prediction error of IER is bounded by three main components: an estimation error $\epsilon_t^{est}$ due to the noise in estimating the $Q$-values $\widehat{Q}_{t,i}^\star$ at the anchor points, an approximation error $(1 + C_t)\epsilon_t^{app}$ due to the linear model defined by the features $\phi_t$, and a propagation error $C_t\epsilon_{t+1}^{bias}$ due to the prediction error of $\widehat{V}_{t+1}^\star$ at timestep $t + 1$. The key result from this lemma is to illustrate how $C_t$ not only impacts the approximation error as in Lem. 1, but it determines how the errors of $\widehat{V}_{t+1}^\star$ propagates from timestep $t + 1$ to $t$. While for a standard least-square method, $C_t$ may be much larger than one, the approximator (2) with the interpolation vector obtained from (3) aims at minimizing the extrapolation and lowering $C_t$ as much as possible, while preserving the linearity of the representation. As discussed in Sect. 4, a suitable choice of the anchor points may significantly reduce the amplification factor by leveraging the additional degrees of freedom offered by choosing $K_t$ larger than $d$. In general, we may expect that the larger $K_t$, the smaller $C_t$. Nonetheless, the overall sample complexity of IER increases as $K_t n_{\text{supp}}$. This shows the need of trading off the number of anchor points (hence possibly higher variance) in exchange for better control on how errors gets amplified. In this sense, Lem. 2 reveals a critical *extrapolation-variance trade-off*.

### 3.2 Sample Complexity of LAVIER

We analyze LAVIER (*Linear Approximate Value Iteration with Extrapolation Reduction*) obtained by running IER backward from timestep $H$ to 1 and we derive a sample complexity upper bound to achieve a near-optimal policy. Under the assumption of bounded value function $V_t^\star(s) \in [0, 1]$ and bounded immediate reward random variables $R(s,a) \in [0, 1]$, we obtain the following result[1]

**Theorem 1.** *Let $C_t \leq C$ and $\epsilon_t^{app} \leq \epsilon^{app}$ for all $t = 1, \ldots, H$. If LAVIER is run with failure probability $\delta > 0$, precision $\overline{\epsilon} > 0$ and constant $\overline{C} > C^H$, $n_{tot} \geq KH^5\overline{C}^2 \ln(2KH/\delta)/\overline{\epsilon}^2$ samples, then with probability $1 - \delta$ LAVIER returns a policy $\widehat{\pi}^\star$ such that*

$$V_1^\star(s_0) - V_1^{\widehat{\pi}^\star}(s_0) \leq \underbrace{\overline{\epsilon}}_{\text{est. error}} + \underbrace{4H^2\overline{C}\epsilon^{app}}_{\text{app. error}}. \tag{8}$$

This bound decomposes the prediction error in two components: an estimation error due to the noise in the samples and an approximation error due to the features $\{\phi_t\}_t$ and the target functions $\{Q_t^\star\}_t$. Thm. 1 illustrates the impact of the amplification factor on the overall sample complexity and final error. If $C > 1$, $\overline{C}$ grows exponentially with the horizon. Furthermore, the error $\epsilon^{app}$ itself is amplified by $\overline{C}$, thus leading to an approx-

---

**Algorithm 1** LAVIER algorithm.

---

**Input**: Failure probability $\delta$, accuracy $\overline{\epsilon}$, set of anchor points $\{\mathcal{K}_t\}_{t=1,\ldots,H}$, time horizon $H$, total amplification constant $\overline{C}$.
Set $\delta' = \delta/(\sum_{t=1}^H K_t)$, $n_{\text{supp}} = \left\lceil \frac{H^4\overline{C}^2}{\overline{\epsilon}^2} \ln(2(\sum_{t=1}^H K_t)/\delta) \right\rceil$
$\widehat{Q}_{H+1}^\star(\cdot) = 0$ (zero predictor at terminal states)
**for** $t = H$ **downto** 1 **do**
    Call IER with param. $(n_{\text{supp}}, \mathcal{K}_t, \widehat{Q}_{t+1}^\star(\cdot))$ and obtain $\widehat{Q}_t^\star(\cdot)$
**end for**
**Return** policy $\widehat{\pi}_t^\star(s) = \arg\max_{a \in \mathcal{A}_{t,s}} \widehat{Q}_t^\star(\phi_t(s,a))$

---

imation error scaling exponentially with $H$. This result is not unexpected, as it confirms previous negative results showing how the extrapolation typical of linear models may lead the error to diverge over iterations [Bai95; TV97]. Nonetheless, if the amplification constant is $C < (1 + \frac{1}{H})$, then $\overline{C} \leq (1 + \frac{1}{H})^H \leq e$, which gives a polynomial sample complexity bound of order $\tilde{O}(KH^5/\overline{\epsilon}^2)$ and a final error where the approximation error is only amplified by $H^2$. While this configuration does remove the divergence problem, it may still lead to a sample inefficient algorithm. In fact, in order to achieve $C \approx 1$, we may need to take $K$ very large. This raises the fundamental question of whether low amplification error and low sample complexity can be obtained at the same time. In the next section, we first discuss how anchor points with small amplification $C$ can be efficiently constructed, while in Sect. 5 we empirically show how in some scenarios this can be achieved with a small number of anchor points $K$ and thus low sample complexity. Finally, we notice that when the features are chosen to be averagers, the interpolation scheme corresponds to a convex combination of anchor weights, thus corresponding to $C = 1$. As a result, Thm. 1 is also a sample complexity result for averagers [Gor95].

## 4 Anchor Points and Amplification Factor

While averagers attain $C = 1$, in general they may not generalize as well as linear models. Furthermore, averagers usually have poor sample complexity, as they may require a number of samples scaling exponentially with the dimension of the state-action space [see e.g., Thm.3 in OS02]. The aim of the minimization program (3) is to trade off the generalization capacity of linear models and their extrapolation, without compromising the overall sample complexity. The process of constructing a "good" set of anchor points can be seen as a form of "experimental design". While in experimental optimal design the objective is to find a small number of anchor points such that least-squares achieves small prediction error, here the objective is to construct a set $\mathcal{K}_t$ such that the amplification factor $C_t$ is small. We have the following result.

**Proposition 1.** *Let $\Phi(\mathcal{K}_t) = \{\phi \in \mathbb{R}^d, \forall (s_i, a_i) \in \mathcal{K}_t, \phi(s_i, a_i) = \phi\}$ be the image of the anchor points through $\phi$. If the convex hull of $\Phi(\mathcal{K}_t)$ contains all the features in $\Phi_t$, i.e.,*

$$\Phi_t \subseteq conv\big(\Phi(\mathcal{K}_t)\big) = \{\phi \in \mathbb{R}^d : \exists \theta^\phi \in \mathbb{R}^{K_t}, \phi = \sum_{i=1}^{K_t} \theta_i^\phi \phi_i, \text{ with } \theta_i^\phi \geq 0, \sum_{i=1}^{K_t} \theta_i^\phi = 1\},$$

*then the amplification factor is $C_t \leq 1$.*

Under the condition of Prop. 1, prediction errors propagates linearly through timesteps. In general, it is not possible to provide a bound on $K_t$, as the number of anchor points needed to construct a convex hull containing $\Phi_t$ may largely vary depending on the structure of $\Phi_t$.[2] If the convex hull is not known

or it contains too many features, an approximate convex hull could be found by standard techniques, for example [GO17; Blu+17] or [SV16; HA13] and can still provably yield a linear propagation of the error if it is of sufficient quality (i.e., $C_t < (1 + 1/H)$). Importantly, finding an approximate convex hull can be performed offline without accessing the generative model as it only requires access to the mapping function $\phi_t(\cdot, \cdot)$. Finally, as the algorithm solves the optimization program (3) during the learning phase (to compute the backup $\widehat{V}_{t+1}^\star(s')$ with the sampled next state $s'$) the actual value of $\|\theta^{\phi(s,a)}\|_1$ is computed and therefore the algorithm can identify whether significant extrapolation is taking place and whether the set of anchor points $\mathcal{K}_t$ may need to be increased or adjusted. While we defer adaptive construction of approximate convex hulls as future work, we propose a simple greedy heuristic to construct a good set of anchor points *before* the learning process.

Let $C$ be a target amplification error, at timestep $t$ we would like to find the smallest set $\mathcal{K}_t$ such that $C(\mathcal{K}_t) = \max_{s,a} \|\theta_t^{\phi_t(s,a)}\|_1$ is below $C$, where the interpolation vector $\theta_t^{\phi_t(s,a)}$ is computed as in (3). As this problem may be NP-hard, we propose a sequential greedy scheme where anchor points are added to $\mathcal{K}_t$ until the condition is met. Starting with $\mathcal{K}_t$ including a single arbitrary state-action $(s_1, a_1)$, if $C(\mathcal{K}_t) > C$, we compute $(\overline{s}, \overline{a}) = \arg\max_{s,a} \|\theta_t^{\phi_t(s,a)}\|_1$ and add it to $\mathcal{K}_t$. Notice that this process does not necessarily return a positive interpolation vector $\theta_i^{\phi_t(s,a)}$ and thus $\widehat{Q}_t^\star$ may not be a convex combination of the anchor values. This extra degree of freedom w.r.t. convex hulls may allow us to obtain a small amplification factor with fewer anchor points. Although we do not have theoretical guarantees about the number of anchor points $K = |\mathcal{K}_t|$ added through this heuristic process, we report experiments where we show that it is possible to effectively obtain small $C$, and thus small prediction error, with few anchor points.

## 5 Numerical Simulations

We investigate the potential benefit of LAVIER over least-squares AVI (LS-AVI). Although LAVIER shares similarity with averagers, a fair comparison is difficult and out of the scope of this preliminary empirical study. In fact, in designing an averager, the choice of structure and parameters (e.g., the position of the points in a nearest neighbor procedure) heavily affects the corresponding function class, i.e., the type of functions that can be accurately represented. As a result, any difference in performance would mostly depend on the different function class used by the averager and the linear model (i.e., $\phi$) used by LAVIER.

The following MDPs are toy examples designed to investigate the differences between the LAVIER and LS-AVI and confirm our theoretical findings. The empirical results are obtained by averaging 100 simulations and they are reported with 95%-confidence intervals.

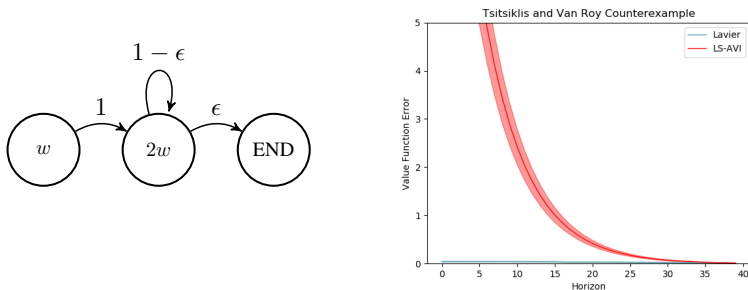

Figure 1: *Left*: Two state MDP. *Right:* Prediction error for least-squares AVI and LAVIER.

**Two-state MDP of Tsitsiklis and Van Roy** The first experiment focuses on how the interpolation scheme of IER may avoid divergence. The smallest-known problem where least-squares approximation diverges is reported in [TV96; SB18]. This problem consists of a two-state Markov reward process (i.e., an MDP with only one action per state) plus a terminal state (Fig. 1). As there is only one possible policy, the approximation problem reduces to estimating its value function. The feature $\phi$ maps a state to a fixed real number, i.e., $\phi(\cdot, \cdot) \in \mathbb{R}$, and there is only one weight to learn. For simplicity, we set the parameter $\epsilon = 0.01$, and add a zero-mean noise to all rewards generated as $1/2 - \text{Ber}(1/2)$, where $\text{Ber}(\cdot)$ is a Bernoulli random variable. We study the approximation error at the left-most state when each algorithm is run for a varying number of iterations $H$ and with 1000

samples at each timestep. The samples are generated uniformly from the left and middle node, which serve as anchor points. Fig. 1 shows that the error of the least-square-based method rapidly diverges through iterations, while LAVIER is more robust and its error remains stable.

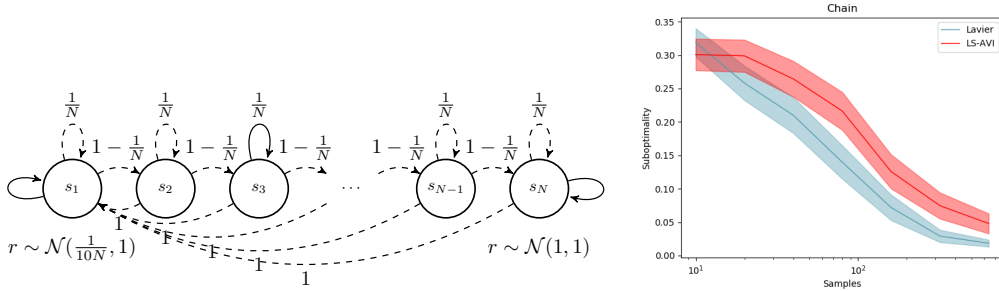

Figure 2: *Left:* Chain MDP. *Right:* Suboptimality of the policy at $s_1$, $V^\star(s_1) - V^{\tilde\pi^\star}(s_1)$.

**Chain MDP.** We now evaluate the quality of the anchor points returned by the heuristic method illustrated in Sect. 4. In the chain MDPs of Fig. 2 the agent starts in the leftmost state and the optimal policy is to always go right and catch the noisy reward in the rightmost state before the episode terminate. However, a small reward is present in the leftmost state and settling for this reward yields a suboptimal policy. We define the feature $\phi(s,a) = [Q^\star(s,a), v(s,a)]$, where $v(s,a) \sim \text{Unif}(0,1)$ is a random number fixed for each simulation and $(s,a)$ pair. We run LS-AVI by sampling state-actions in the reachable space uniformly at random, while for LAVIER we compute an anchor set with $C \le 1.2$. Both algorithms use the same number of samples and LAVIER splits the budget of samples uniformly over the anchor points to compute the anchor values. The length of the chain is $N = 50$, which is also the time horizon. We report the quality of the learned policy at $s_1$ and notice that LAVIER is consistently better than LS-AVI (see App. A for further experiments).

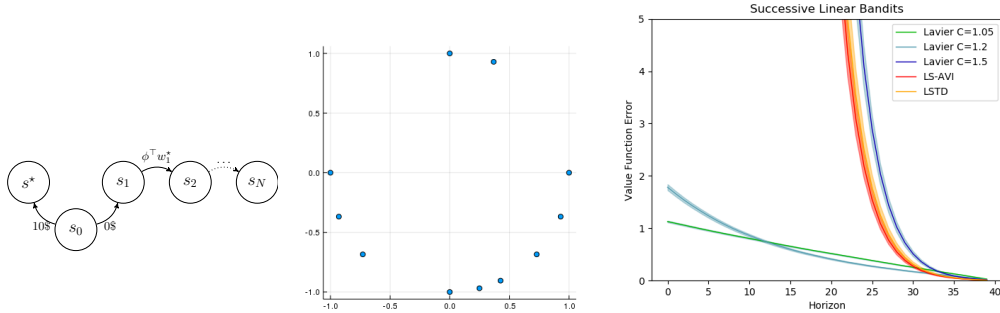

Figure 3: *Left:* MDP with a sequence of linear bandits with actions in 2 dimensions. *Center:* Example of the anchor points generated by the heuristic greedy algorithm. *Right:* Accuracy $V^\star(s_t) - V^{\tilde\pi^\star}(s_t)$ as a function of state.

**Successive Linear Bandits.** We consider an MDP defined as a sequence of linear bandit problems (Fig. 3) which is designed so that significant extrapolation occurs at each iteration. In this MDP, there are $N$ states $s_1, \ldots, s_N$ augmented with the starting state $s_0$ and a terminal state $s^\star$. From the starting state $s_0$ there are two actions (*left* and *right*). The optimal policy is to take *left* and receive a reward of 10. The states $s_1, \ldots, s_N$ are linear bandit problems, where each action gives a Gaussian noisy return of mean 0 and variance 1 and the state transitions deterministically from $s$ to $s+1$. This represents a sequence of linear bandits with no signal, i.e., the output is not correlated with the features and the learner only experiences noise, hence $V_1^\star(s_1) = 0$. The feature map $\phi_t(s,a) = \phi_a$ returns the features describing the action itself, and the solution $Q_t^\star(s,a) = 0$ is exactly representable by a zero weight vector. The solution is unique. The learner should estimate the value of $V_t^\star(s_1)$ accurately to infer the right action in state $s_0$. At each state $s_1, \ldots, s_N$, we represent actions in $\mathbb{R}^2$ and we generate 100 actions by uniformly discretizing the circumference. As the canonical vectors $e_1$ and $e_2$ are the most informative actions to estimate the reward associated to any other action

(see [SLM14] for the best policy identification in linear bandits), we collect our samples from these two actions. The anchor points for LAVIER are chosen by our adaptive procedure for different value of the extrapolation coefficient $C \in \{1.05, 1.2, 1.5\}$. The extrapolation becomes more and more controlled as $C$ approaches one. Fig 3 shows the performance at different states. For small values of $C$, LAVIER significantly outperforms LS-AVI. Furthermore, looking more closely into the rightmost states (i.e., the states that are updated at early iterations) reveals the extrapolation-variance tradeoff (see Fig. 5 in App. B for a zoomed version of the plot): a value of $C = 1.5$ ensures a more accurate estimate (due to less variance) in the first timesteps, but the curve steeply diverges. By contrast, $C = 1.05$ has initially a poorer estimate, but such estimate remains far more stable with the horizon. We also report the support points selected by the algorithm. Although $C$ is small, only a few points are necessary. In fact, we do not need to cover the circle with an approximate convex hull and our procedures can, for example, 'flip' the sign of the learned value without causing extrapolation (i.e., keeping $C$ small).

In Fig. 3 we also report the performance of LS-AVI. In this case, the divergence of the estimate of LS-AVI is extreme, and it does not allow to accurately estimate $V^\star(s_1) = 0$, yielding a policy that cannot identify the correct initial action. Furthermore, in this example we additionally evaluate Least Square Temporal Difference (LSTD) for off-policy prediction [SB18]. LSTD is not a policy optimization algorithm but we can use it to evaluate the value of a policy that chooses for example the action $[1/\sqrt{2}, 1/\sqrt{2}]$ in every state of the chain. The training data for LSTD are identical to LS-AVI, i.e., the canonical vectors $e_1$ and $e_2$. Despite collecting data along the informative direction $e_1$ and $e_2$, the LSTD solution is of increasingly poor quality as a function of the chain length.

## 6 Conclusion

**Related work.** Most of literature in linear function approximation focused on designing feature maps $\phi$ that could represent action-value functions well, by optimizing parameterized features (e.g., in deep networks or in [MMS05]), by an initial representation learning phase to extract features adapted to the structure of the MDP [MM07; Pet07; Bel+19], or by adding features to reduce the approximation error [Tos+17]. Unfortunately, accurately fitting value functions does not guarantee small inherent Bellman error (IBE), and thus LAVI may still be very unstable. In this paper we assume $\phi$ has small approximation error but arbitrary IBE and we focus on how to reduce the amplification factor at each iteration.

Yang and Wang [YW19] recently studied the sample complexity of LAVI under the assumption that the transition model $p$ admits a non-negative low-rank factorization in the features $\phi$. In particular, they show that in this case, the inherent Bellman error is zero, thus avoiding the amplification of errors through iterations of LAVI. In this paper, we consider the more general case where only the optimal action-value function should be accurately approximated in $\phi$, which may be true even when the transition model does not admit a low-rank decomposition. In fact, Thm. 1 holds even when the inherent Bellman error is infinite and shows that whenever the amplification factor $C$ is small, LAVIER can still achieve polynomial sample complexity. Yang and Wang [YW19] used the convexity condition in Prop. 1 to derive sample complexity guarantees in their setting, while in our case, the same condition is used to control the amplification of errors. Furthermore, we notice that in LAVIER we only need to control the $\ell_1$-norm of the interpolation weights, which does not necessarily require any convexity assumption (see also experiments). [YW19] introduced OPPQ-Learning and proved a near-optimal sample complexity bound of order $\widetilde{O}(K/\epsilon^2(1-\gamma)^3)$, where $K$ is the number of anchor points which are assumed to be provided[3]. While this shows that both methods scale linearly with the number of anchor points, OPPQ-Learning enjoys a much better dependency on the horizon. It remains as an open question whether our analysis for LAVIER can be improved to match their bound or the difference the unavoidable price to pay for the more general setting we consider[4].

Averagers pursue the same objective but take an extreme approach, where no extrapolation is allowed and Q-functions are approximated by interpolation of values at a fixed set of anchor points [Gor95; Gor96; PP13; KKL03; MM99]. Unfortunately, such approach may suffer from a poor sample complexity [PP13; KKL03], as the number of anchor points may scale exponentially with the

problem dimensionality. In LAVIER, we introduce a more explicit extrapolation-variance tradeoff, where the anchor points should be designed to avoid extrapolation only when/where it happens.

**Future work.** There are several directions for future investigation. AVI is a core building block for many exploration-exploitation algorithms [OVW16, Kum+18] and better LAVI may help in building sample-efficient online learning algorithms with function approximation. Another venue of investigation is off-policy prediction with batch data. The mismatch between behavioural and target policies poses similar challenges as in the error propagation of AVI. In order to control the extrapolation-variance tradeoff may need penalize a non-uniform use of the samples (to reduce the variance) while the 1-norm minimization objective may reduce the amount of extrapolation to the desired value.

## Acknowledgment

This was was partially supported by a Total Innovation Fellowship. The authors are grateful to the reviewers for the high quality reviews and helpful suggestions.

## Footnotes

[1]This assumption is inspired by [JA18], who suggested this is a more expressive framework, as it allows some rewards to be substantially larger than others in terms of contributing to the final value function.

[2]For instance, if $\Phi_t$ is a polyhedron in $\mathbb{R}^d$, $K_t$ may be as large as exponential in $d$.

[3]Yang and Wang [YW19] point out that convex assumption requires the number of features $d$ to scale with the number of anchor points $K$.

[4]We conjecture that the dependency on $H$ could be greatly improving using similar arguments as in [YW19], such as monotonicity, tighter concentration inequalities, and variance reduction.

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
