[Supplementary Material · neurips_leavi.pdf]

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

$$\widehat{Q}_t^{\star}(s, a) = \sum_{i=1}^{K_t} \theta_{t,i}^{\phi_t(s,a)} \widehat{Q}_{t,i}^{\star}, \tag{2}$$

where the interpolation vector $\theta_t^{\phi_t(s,a)} \in \mathbb{R}^K$ is the solution to the optimization problem

$$\min_{\theta^{\phi_t(s,a)}} \|\theta^{\phi_t(s,a)}\|_1 \qquad \text{subject to} \quad \phi_t(s, a) = \sum_{i=1}^{K_t} \theta_i^{\phi_t(s,a)} \phi_t(s_i, a_i). \tag{3}$$

As long as the image of the anchor points $\{\phi(s_i, a_i)\}_{i=1}^{K_t}$ spans $\mathbb{R}^d$, (3) admits a solution. This problem is a *linear* optimization program with *linear* constraints and it can be solved efficiently using standard techniques [BV04; NW06]. Notice that the weights $\theta_t^{\phi_t(s,a)}$ change with $s, a$ and no positiveness constraint is enforced.

## 3.1 Prediction Error and Sample Complexity of IER

In most problems, the optimal action-value function $Q_t^{\star}$ cannot be exactly represented by a low dimensional inner product $w_t^{\top} \phi_t(\cdot, \cdot)$. The best approximator that can be expressed by features $\phi$ and its associated approximation error are defined as

$$w_t^{\star} = \arg\min_{w \in \mathbb{R}^d} \|w^{\top} \phi_t(\cdot) - Q_t^{\star}(\cdot)\|_{\infty}; \qquad \epsilon_t^{app} = \min_{w \in \mathbb{R}^d} \|w^{\top} \phi_t(\cdot) - Q_t^{\star}(\cdot)\|_{\

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

# A    Additional Experiments on Chain MDP

Figure 4: Chain MDP with $N = 25, 100$ and shifting sampling distribution.

We consider the chain MDP illustrated in the main text. In this additional experiment we illustrate how LAVIER may be more robust than LS-AVI when off-policy distributions are used to collect samples. We set $\phi_t(s, a) = Q_t^\star(s, a)$. This way there is no approximation error as the algorithm learns one weight for each timestep (there is a total of $N$ parameters to learn). In this experiment samples are collected at $s = t$ for both the "left" and "right" action which also represent the anchor points for LAVIER (i.e., the anchor set is not optimized for small amplification). Notice that even collecting samples exclusively from, say, action "left" is enough to perfectly reconstruct the optimal action-value function, as it is enough to set $w_t = 1$ to obtain $\widehat{Q}_t^\star = Q_t^\star$. Nonetheless, the sampling distribution may still affect how the estimation errors are propagated through iterations. In Fig. 2 we examine the effect of progressively shifting the sampling distribution away from the distribution of the optimal policy. We define the ratio:

$$\text{ratio} = \frac{\text{\# samples to "right" action}}{\text{\# samples to "left" action}}.$$

When the ratio is small, the sampling distribution favors the "left" action, which is not what the optimal policy chooses in most state-timesteps pairs. The plots in Fig. 2 show that LS-AVI and LAVIER perform similarly when the ratio is 1. On the other hand, as the ratio favors the "left" action the performance degrades, but much less so for LAVIER, which remains accurate even under a severe shifts in the distribution.

# B    Additional Results for Sequence of Linear Bandits

Figure 5: Detailed view of Fig. 3.

Fig. 5 reports a more detailed view of the performance in the rightmost states.

We also test the same setting but using a three dimensional action space. Same setting as described in the main text; LS-AVI and LSTD learns along the coordinate axes $e_1, e_2, e_3$. The action set is obtained by discretizing the surface of the sphere by 11 points using polar cordinate (11 points for each angle), and the support points are a subset of these, discretized by 3 points (for each angle).

## C    Technical Tools

In this section we recall basic technical tools used in the proof of our main results.

**Proposition 2** (Hoeffding's Inequality [Hoe63]). *Let $X_1, \ldots, X_n$ be i.i.d. random variables with values in $[0, 1]$ and let $\delta' > 0$. Then with probability at least $1 - \delta'$ in $(X_1, \ldots, X_n)$ we have:*

$$\left| \mathbb{E} X - \frac{1}{n} \sum_{i=1}^{n} X_i \right| \leq \sqrt{\frac{\ln(2/\delta')}{2n}} \tag{9}$$

**Proposition 3** (Simulation Lemma, Change of Policy). *For any two fixed policies policies $\pi_1, \pi_2$ on MDP $\mathcal{M}$ and timestep $\tau \in [H]$ it holds that:*

$$V_\tau^{\pi_1}(s_0) - V_\tau^{\pi_2}(s_0) = \mathbb{E}_{\pi_2, s_0} \left[ \sum_{t=\tau}^{H} \left( Q_t^{\pi_1}(s, \pi_1(s, t)) - Q_t^{\pi_1}(s, \pi_2(s, t)) \right) \right]. \tag{10}$$

*Proof.*

$$V_\tau^{\pi_1}(s_0) - V_\tau^{\pi_2}(s_0)$$
$$= r(s, \pi_{1,\tau}(s_0)) - r(s, \pi_{2,\tau}(s_0)) + p(s, \pi_{1,\tau}(s_0))^\top V_{\tau+1}^{\pi_1} - p(s, \pi_{2,\tau}(s_0))^\top V_{\tau+1}^{\pi_2}$$
$$= r(s, \pi_{1,\tau}(s_0)) - r(s, \pi_{2,\tau}(s_0)) + p(s, \pi_{1,\tau}(s_0))^\top V_{\tau+1}^{\pi_1} - p(s, \pi_{2,\tau}(s_0))^\top V_{\tau+1}^{\pi_1} +$$
$$\quad + p(s, \pi_{2,\tau}(s_0))^\top \left( V_{\tau+1}^{\pi_1} - V_{\tau+1}^{\pi_2} \right)$$
$$= Q_\tau^{\pi_1}(s, \pi_{1,\tau}(s_0)) - Q_\tau^{\pi_1}(s, \pi_{2,\tau}(s_0)) + \mathbb{E}_{\pi_2, s_0} \left[ V_{\tau+1}^{\pi_1}(s) - V_{\tau+1}^{\pi_2}(s) \right].$$

Induction concludes the proof.    □

## D    Properties of Problem (3) and Proof of Lemma 1

In order to simplify the notation, in this section we define $\mathcal{X} = \mathcal{S} \times \mathcal{A}$, so that $x = (s, a)$ and we drop the dependency on $t$, as the same reasoning applies at any timestep. For instance, an action-value function $Q^\star$ evaluated at $x_i = (s_i, a_i)$ is denoted by $Q^\star(x_i)$.

We first report this simple result about the optimization problem in (3).

**Lemma 3.** *Let $\mathcal{K}$ be the set of anchor points and $\Phi(\mathcal{K}) = \{\phi \in \mathbb{R}^d, \forall x_i \in \mathcal{K}, \phi(x_i) = \phi\}$ the image of $\mathcal{K}$ through the feature map $\phi : \mathcal{X} \to \mathbb{R}^d$. If the cardinality of the $d$-dimensional anchor features is at least $d$ and these are generators for $\mathbb{R}^d$ then the minimization problem of (3) admits a solution.*

*Proof.* From at least $d$ generators a basis for $\mathbb{R}^d$ with $d$ generators can be extracted. Without loss of generality, assume that this basis consists of the first $d$ features $\{\phi_1, \ldots, \phi_d\}$. This implies that there exists coefficients $[\theta_1^{\phi(x)}, \ldots, \theta_d^{\phi(x)}]$ such that $\phi(x) = \sum_{i=1}^{d} \theta_i^{\phi(x)} \phi(x_i)$ is satisfied. Therefore, by setting to zero the remaining coefficients we obtain a feasible solution $[\theta_1^{\phi(x)}, \ldots, \theta_d^{\phi(x)}, 0, \ldots, 0]$ for the optimization problem (3). Since the optimization problem is bounded below, a minimizer must exist.    □

We now study the properties of the "exact" version of IER. Since IER is a generic approximation algorithm, this analysis holds for an regression task. Given a target function $Q^\star : \mathcal{X} \to \mathbb{R}$ and a feature map $\phi : \mathcal{X} \to \mathbb{R}^d$, we recall that the best linear approximation of $Q^\star$ and its corresponding $\ell_\infty$-error are

$$w^\star = \arg\min_{w \in \mathbb{R}^d} \max_{x \in \mathcal{X}} \left| \phi(x)^\top w - Q^\star(x) \right|, \qquad \epsilon = \max_{x \in \mathcal{X}} \left| \phi(x)^\top w^\star - Q^\star(x) \right|. \tag{11}$$

Given a set of anchor points $\mathcal{K}$, we study the approximator

$$\widetilde{Q}^\star(x) \overset{def}{=} \sum_{i=1}^{K} \theta_i^{\phi(x)} Q^\star(x_i), \tag{12}$$

where the interpolation coefficients $\theta^{\phi(x)}$ are obtained by solving (3). We prove Lemma 1.

*Proof.* First, we show that $\widetilde{Q}^\star(x)$ can still represent linear functions. Let $\epsilon = 0$, i.e., $Q^\star(x) = \phi(x)^\top w^\star$, then we have

$$\widetilde{Q}^\star(x) = \Big( \sum_{i=1}^{K} \theta_i^{\phi(x)} \phi(x_i)^\top \Big) w^\star = \phi(x)^\top w^\star, \tag{13}$$

where we used the constraint in the definition of $\theta^{\phi(x)}$.

We now move to studying the amplification of error $\epsilon$. By adding and subtracting the best linear model $\phi^\top w^\star$:

$$\max_{x \in \mathcal{X}} \left| \widetilde{Q}^\star(x) - Q^\star(x) \right| = \max_{x \in \mathcal{X}} \left| \widetilde{Q}^\star(x) - \phi(x)^\top w^\star + \phi(x)^\top w^\star - Q^\star(x) \right| \tag{14}$$

Next, we use the triangle inequality with the fact that $\max_x a(x) + b(x) \le \max_x a(x) + \max_x b(x)$ to obtain:

$$\max_{x \in \mathcal{X}} \left| \widetilde{Q}^\star(x) - Q^\star(x) \right| \le \max_{x \in \mathcal{X}} \left| \widetilde{Q}^\star(x) - \phi(x)^\top w^\star \right| + \underbrace{\max_{x \in \mathcal{X}} \left| \phi(x)^\top w^\star - Q^\star(x) \right|}_{\epsilon}. \tag{15}$$

Next, using the constraint in the optimization problem used to compute the interpolation coefficients of $\widetilde{Q}^\star(x)$, we have

$$\max_{x \in \mathcal{X}} \left| \widetilde{Q}^\star(x) - Q^\star(x) \right| \le \max_{x \in \mathcal{X}} \left| \sum_{i=1}^{K} \theta_i^{\phi(x)} Q^\star(x_i) - \Big( \sum_{i=1}^{K_t} \theta_i^{\phi(x)} \phi(x_i) \Big)^\top w^\star \right| + \epsilon \tag{16}$$

$$= \max_{x \in \mathcal{X}} \left| \sum_{i=1}^{K_t} \theta_i^{\phi(x)} \big( Q^\star(x_i) - \phi(x_i)^\top w^\star \big) \right| + \epsilon \tag{17}$$

where the second equality is just grouping the factors. The triangles inequality, the definition of best linear model and of amplification constant justifies the upper bound below, which is the statement,

$$\max_{x \in \mathcal{X}} \left| \widetilde{Q}^\star(x) - Q^\star(x) \right| \le \max_{x \in \mathcal{X}} \sum_{i=1}^{K_t} \left| \theta_i^{\phi(x)} \right| \underbrace{\left| Q^\star(x_i) - \phi(x_i)^\top w^\star \right|}_{\le \epsilon} + \epsilon \le (C+1)\epsilon. \tag{18}$$

$\square$

# E    Proof of Lemma 2

*Proof.* If we indicate with $R^{(j)}$ and $s^{(j)}$ the $j$-th sample of the reward and successor state at timestep $t$ at the support point $(s_i, a_i)$, Hoeffding inequality ensures:

$$\left| \frac{1}{n_{\text{supp}}} \Big( \sum_{j=1}^{n_{\text{supp}}} R^{(j)} + Q^\star_{t+1}(s^{(j)}) \Big) - V^\star_t(s_i, a_i) \right| \le \sqrt{\frac{\ln (2/\delta')}{2n_{\text{supp}}}} \overset{def}{=} \epsilon_t^{est} \tag{19}$$

since by assumption $0 \leq Q_t^\star(s,a) \leq V_t^\star(s) \in [0,1]$, $\forall (s,a) \in \mathcal{S} \times \mathcal{A}$ with probability at least $1 - \delta'$. If we set $\delta' \stackrel{def}{=} \frac{\delta}{KH}$, then a union bound over the $K$ anchor points at level $t$ ensures the above holds true for all anchor points jointly with probability at least $1 - \frac{\delta}{H}$ as long as

$$n_{\text{supp}} \geq \frac{\ln(2HK/\delta)}{4(\epsilon_t^{est})^2}. \tag{20}$$

We denote with $\widehat{r}_t(s_i, a_i)$ and $\widehat{\mathbb{E}}_{s'}^{i,t} \max_{a \in \mathcal{A}_{t,s'}} \widehat{Q}_{t+1}^\star(s', a)$ the empirical estimate of the reward and expected Bellman backup using the next-state empirical predictor $\widehat{Q}_{t+1}^\star$, i.e.,

$$\widehat{Q}_{t,i}^\star = \widehat{r}_t(s_i, a_i) + \widehat{\mathbb{E}}_{s'}^{i,t} \max_{a \in \mathcal{A}_{t,s'}} \widehat{Q}_{t+1}^\star(s', a),$$

where the superscript on the expectation recalls its dependency on the anchor point $i$ at timestep $t$. Let $\widehat{\epsilon}_t(s,a) = |\widehat{Q}_t^\star(s,a) - Q_t^\star(s,a)|$, then by using the definition of $\widehat{Q}_t^\star$ we have

$$\widehat{\epsilon}_t(s,a) = \left| \sum_{i=1}^K \theta_i^{\phi(x)} \left( \widehat{r}_t(s_i, a_i) + \widehat{\mathbb{E}}_{s'}^{(s_i,a_i,t)} \max_{a \in \mathcal{A}} \widehat{Q}_{t+1}^\star(s', a) \right) - Q_t^\star(s,a) \right|$$

Next, we add and subtract the empirical expectation of the next-state *true* $Q$ value function $\widehat{\mathbb{E}}_{s'}^{i,t} \max_{a \in \mathcal{A}} Q_{t+1}^\star(s', a)$ to obtain:

$$
\begin{aligned}
\widehat{\epsilon}_t(s,a) = & \left| \sum_{i=1}^K \theta_i^{\phi(x)} \left( \widehat{r}_t(s_i, a_i) \right. \right. \\
& \left. \left. + \widehat{\mathbb{E}}_{s'}^{i,t} \left( \max_{a \in \mathcal{A}} \widehat{Q}_{t+1}^\star(s', a) - \max_{a \in \mathcal{A}_{t,s'}} Q_{t+1}^\star(s', a) + \max_{a \in \mathcal{A}} Q_{t+1}^\star(s', a) \right) - Q_t^\star(s,a) \right) \right| \\
= & \left| \sum_{i=1}^K \theta_i^{\phi(x)} \left( \widehat{r}_t(s_i, a_i) + \widehat{\mathbb{E}}_{s'}^{i,t} \max_{a \in \mathcal{A}} Q_{t+1}^\star(s', a) - Q_t^\star(s_i, a_i) \right) \right. \\
& + \left( \sum_{i=1}^K \theta_i^{\phi(x)} Q_t^\star(s_i, a_i) - Q_t^\star(s,a) \right) \\
& \left. + \sum_{i=1}^K \theta_i^{\phi(x)} \left( \max_{a \in \mathcal{A}} \widehat{Q}_{t+1}^\star(s', a) - \max_{a \in \mathcal{A}} Q_{t+1}^\star(s', a) \right) \right|
\end{aligned}
$$

The triangle inequality allows us to upper bound the above expression with the one below:

$$
\begin{aligned}
\widehat{\epsilon}_t(s,a) \leq & \sum_{i=1}^K |\theta_i^{\phi(x)}| \left| \widehat{r}_t(s_i, a_i) + \widehat{\mathbb{E}}_{s'}^{i,t} \max_{a \in \mathcal{A}_{t,s'}} Q_{t+1}^\star(s', a) - Q_t^\star(s_i, a_i) \right| \\
& + \left| \sum_{i=1}^K \theta_i^{\phi(x)} Q_t^\star(s_i, a_i) - Q_t^\star(s,a) \right| \\
& + \sum_{i=1}^K |\theta_i^{\phi(x)}| \left| \max_{a \in \mathcal{A}} \widehat{Q}_{t+1}^\star(s', a) - \max_{a \in \mathcal{A}_{t,s'}} Q_{t+1}^\star(s', a) \right|.
\end{aligned}
$$

Finally, under the event that (19) holds true for all $K$ anchor points then $\left| \widehat{r}_t(s_i, a_i) + \widehat{\mathbb{E}}_{s',\phi_{i,t}} V_{t+1}^\star(s') - Q_t^\star(s_i, a_i) \right| \leq \epsilon_t^{est}$ holds true; together with the assumption

on the approximation at the next timestep and Lemma 1 the upper bound below is justified

$$\widehat{\epsilon}_t(s,a) \leq \sum_{i=1}^{K} \left| \theta_i^{\phi(x)} \right| \underbrace{\left| \widehat{r}_t(s_i, a_i) + \widehat{\mathbb{E}}_{s'}^{i,t} \max_{a \in \mathcal{A}} Q_{t+1}^{\star}(s', a) - Q_t^{\star}(s_i, a_i) \right|}_{\leq \epsilon_t^{est}}$$

$$+ \underbrace{\left| \sum_{t=1}^{H} \theta_i^{\phi(x)} Q_t^{\star}(s_i, a_i) - Q_t^{\star}(s, a) \right|}_{\leq (C_t + 1)\epsilon_t^{app}}$$

$$+ \sum_{i=1}^{K} \left| \theta_i^{\phi(x)} \right| \underbrace{\left| \max_{a \in \mathcal{A}} \widehat{Q}_{t+1}^{\star}(s', a) - \max_{a \in \mathcal{A}} Q_{t+1}^{\star}(s', a) \right|}_{\leq \epsilon_{t+1}^{bias}}.$$

Finally, the definition of amplification constant justifies the final upper bound

$$\leq C_t \epsilon_t^{est} + (C_t + 1)\epsilon_t^{app} + C_t(\epsilon_{t+1}^{bias}) = \epsilon_t^{app} + C_t(\epsilon_t^{est} + \epsilon_t^{app} + \epsilon_{t+1}^{bias}).$$

$\square$

The above proposition is crucial in that it claims that if next state value function error in the $\ell_\infty$-norm is not too high then we obtain a uniformly good approximation for the $Q^\star$ values at the current timestep, with high probability. From this, we can easily deduce that the optimal value function estimate $\widehat{V}^\star$ for the current estimate is also probably approximately correct in the infinity norm, as we show below.

## F    Proof of Theorem 1

We study how the errors in the optimal value function estimate $\|\widehat{V}_t - V_t^\star\|_\infty$ propagate through different timesteps $t$ with the assumption that we can obtain an uniform (in timesteps) upper bound to the noise and approximation error.

**Lemma 4** (Propagation of Errors). *Let* $\|V_t^\star - \widehat{V}_t^\star\|_\infty \leq \epsilon_t^{bias}$*. If*

$$\epsilon_t^{bias} \leq \epsilon_t^{app} + C_t(\epsilon_t^{est} + \epsilon_t^{app}) + C_t\epsilon_{t+1}^{bias} \tag{21}$$

*holds for all timesteps* $t \in [H]$ *and*

$$C_t \leq C, \quad \forall t \tag{22}$$
$$\epsilon_t^{est} \leq \epsilon^{est}, \quad \forall t \tag{23}$$
$$\epsilon_t^{app} \leq \epsilon^{app}, \quad \forall t \tag{24}$$

*then it holds that:*

$$\epsilon_t^{bias} = (1 + C + \cdots + C^{H-t})(\epsilon^{app} + C(\epsilon^{est} + \epsilon^{app})) \leq H\frac{\overline{C}}{C}(\epsilon^{app} + C(\epsilon^{est} + \epsilon^{app})). \tag{25}$$

*where the problem dependent constant* $\overline{C}$ *is defined as:*

$$\overline{C} = C^H. \tag{26}$$

*Proof.* By assumption,

$$\epsilon_t^{bias} \leq \underbrace{\epsilon^{app} + C(\epsilon^{est} + \epsilon^{app})}_{\overset{def}{=} F} + C\epsilon_{t+1}^{bias} = F + C\epsilon_{t+1}^{bias} \tag{27}$$

The inductive hypothesis is that $\epsilon_{t+1}^{bias} = (1 + C + \cdots + C^{H-t-1})F$ holds. Together with the above statement, we get $\epsilon_t^{bias} \leq (1 + C + \cdots + C^{H-t})F$, which is the statement. $\square$

Under the hypothesis that the value function is accurately estimated we get that the resulting greedy policy (on the inaccurate model) is also near-optimal on the true model.

**Lemma 5** (From Value Function Accuracy to Policy Accuracy). *If at any timestep $t \in H$ it holds that:*

$$\|V_t^\star - \widehat{V}_t^\star\|_\infty \leq \epsilon_t^{bias} \tag{28}$$

*then for any any starting state $s_0$ the policy $\widehat{\pi}^\star$ returned by Algorithm* LAVIER *satisfies:*

$$V_1^\star(s_0) - V_1^{\widehat{\pi}^\star}(s_0) \leq 2H\epsilon_1^{bias} \tag{29}$$

*with probability at least $1 - \delta$.*

*Proof.* The simulation lemma in Prop. 3 yields the following sequence of inequalities, where $\mathbb{E}_{\widehat{\pi}^\star, s_0}$ is the expectation over the trajectories identified by the policy $\widehat{\pi}^\star$ returned by LAVIER upon starting from $s_0$:

$$V_1^\star(s_0) - V_1^{\widehat{\pi}^\star}(s_0) = \mathbb{E}_{\widehat{\pi}^\star, s_0} \sum_{t=1}^H Q_t^\star(s, \pi^\star(s)) - Q_t^\star(s, \widehat{\pi}^\star(s)) \tag{30}$$

$$\overset{(a)}{\leq} \mathbb{E}_{\widehat{\pi}^\star, s_0} \sum_{t=1}^H \widehat{Q}_t^\star(\phi_t(s, \pi^\star(s))) + \epsilon_t^{bias} - \widehat{Q}_t^\star(\phi_t(s, \widehat{\pi}^\star(s))) + \epsilon_t^{bias} \tag{31}$$

$$\overset{(b)}{\leq} \mathbb{E}_{\widehat{\pi}^\star, s_0} \sum_{t=1}^H 2\epsilon_t^{bias} \tag{32}$$

$$\overset{(c)}{\leq} 2\epsilon_1^{bias} \mathbb{E}_{\widehat{\pi}^\star, s_0} \sum_{t=1}^H = 2H\epsilon_1^{bias} \tag{33}$$

Step $(a)$ is justified by the induction step, while step $(b)$ uses the fact that the algorithm always returns an action that in state-timestep $(s, t)$ maximizes the $\widehat{Q}_t^\star(\phi_t(s, a))$ values. Finally, $(c)$ follows from the fact that $\epsilon_t^{bias} \geq \epsilon_{t+1}^{bias}$ for all $t$. $\square$

We now present our main result:

*Proof.* The conditions of Lemma 5 hold with probability at least $1 - \delta$, so that we can bound the performance of policy $\widehat{\pi}^\star$ by requiring

$$\epsilon_1^{bias} \leq \frac{\overline{\epsilon}}{2H} + H\frac{\overline{C}}{C}(C + 1)\epsilon^{app}. \tag{34}$$

$\square$

Provided that we can find an uniform bound on $\epsilon_t^{est}$ for all $t$ to satisfy the hypothesis of lemma 4, we need to set:

$$\epsilon_1^{bias} \leq H\frac{\overline{C}}{C}(\epsilon^{app} + C(\epsilon^{est} + \epsilon^{app})) \leq \frac{\overline{\epsilon}}{2H} + H\frac{\overline{C}}{C}(C + 1)\epsilon^{app} \tag{35}$$

where the fist inequality is given by lemma 4. We can cancel the inapproximability error $H\frac{\overline{C}}{C}(C + 1)\epsilon^{app}$, yielding:

$$H\overline{C}\epsilon^{est} \leq \frac{\overline{\epsilon}}{2H} \tag{36}$$

which readily yields the maximum permissible error on the noise:

$$\epsilon^{est} \leq \frac{\overline{\epsilon}}{2H^2\overline{C}}. \tag{37}$$

This implies that we need to set $n_{\text{supp}}$ to

$$n_{\text{supp}} = \left\lceil \frac{\ln(2/\delta')}{4(\epsilon_t^{est})^2} \right\rceil = \left\lceil \frac{\ln(2/\delta') \times 4H^4\overline{C}^2}{4(\overline{\epsilon})^2} \right\rceil = \left\lceil \frac{H^4\overline{C}^2}{\overline{\epsilon}^2} \ln(2KH/\delta) \right\rceil. \tag{38}$$

Since LAVIER solves $K$ estimation problems at each of the $H$ timesteps, a total complexity bound:

$$n_{tot} = KH \left\lceil \frac{H^4 \overline{C}^2}{\overline{\epsilon}^2} \ln(2KH/\delta) \right\rceil \tag{39}$$

follows.