[Reviews · NeurIPS 2019]

Reviewer 1



Originality The work is in some ways similar to Yang and Wang but takes different approaches and derives different insights. Originality is good. Quality I find the quality very good; the work is thorough and the experimental results are appropriate to their purpose, which is to illustrate the theory rather than to provide state-of-the-art applied results. l.119 "The best approximator..." - There are many definitions of a "best approximator." Please give some insight to the reader on why this particular definition of "the best approximator" was chosen. Throughout: "Lem." and "Thm." are used inconsistently with inconsistent spacing. Just write them out. l.150 - "minimizing ... and lower" - "minimizing ... and lowering" l.298 "represent well action-value functions" - "represent action-value functions well" The paper is very well written. I have a few specific comments. Significance As indicated above, I think the ideas will be influential in future work on limited-data RL problems. *** I have read the author response and I appreciate their clarifications.

Reviewer 2



After rebuttal: Thanks the authors for addressing my concerns. I read the authors' feedback and other reviews. This work has some contribution on the theoretical side but I believe its empirical contribution is limited. The experiment is simple and the algorithm might not work very well in general. I'll keep my original score. ----------------------------------- This paper studies the finite horizon value iteration problem with linear approximation given a generative model. This paper is clearly written and easy to follow. The author proposes an algorithm and then prove a sample complexity bound for the algorithm. Finally, some experiments on the toy models are shown to support the theoretical result. The AVI problem has been studied for a long time. There are mainly two types of guarantees needed for the convergence. One is the bounded inherent Bellman error and the other is the contraction of Bellman update. The main idea of this paper is to balance the generalization and compactness in LAVI. The algorithm will run backward. At timestep t, the agent uses the generative model to draw the backup values and timestep t+1. Then it solves an optimization problem to obtain the interpolation coefficient \theta of each anchor point in timestep t. The sample complexity will be low if the amplification factor and the number of anchor points can be controlled. The result will hold even under infinite inherent Bellman error. Major: 1. The idea of trying to balance the compactness and generalization is new to me. 2. The sample complexity is polynomial on the number of anchor points K, the length of horizon H, and the amplification factor \bar{C}. However, the limitation is that \bar{C} seems usually to be exponential. Although Prop 1 shows a specific case that \bar{C} is small, the number of anchor point seems not controllable. Usually, the smaller the amplification factor, the larger the number of anchor points. My major concern is that the final sample complexity might still be exponential. 3. This paper does not consider the selection of anchor points. I agree that finding the anchor points are generally hard. Suppose a set of (probably good) anchor points are given in the last timestep H, is it possible to design an algorithm to automatically find the anchor points from timestep from H-1 downto 1? I’m also interested that whether the result of this work can be extended to the infinite horizon case since a closely related work Yang and Wang [15] considers discounted infinite horizon MDP. 4. In the theoretical part, the result holds for the continuous state space. However, the authors only investigate the discrete state space in experiment. Does the algorithm also work empirically in the continuous state space? Minor: 1. Line 178: in not unexpected --> is not unexpected 2. Line 252: Fig. 5 --> Fig. 2

Reviewer 3



## Summary This paper consider linear-combination-of-features approximation to the Q-value function in a finite horizon reinforcement learning problem. While the linear approximation requires few parameters and scales efficiently with the state and action space dimension, when Q-valuate iteration is applied to it, the combination of least-squares projection and Bellman operator applications may be expansive leading to divergence. The paper proposes to use the linear approximation to the Q-function at a finite set of anchor states and linearly interpolate to other state-action pairs. This balances the compactness of a linear approximation with reduction in error amplification after Bellman updates due to the interpolation. The authors prove a bound on the error of the proposed Q-function interpolator with respect to the best linear linear-combination-of-features approximation to the Q-value function. ## Recommendation The paper is well written and easy to follow. The idea of trading off linear approximation and extrapolation in linear-combination-of-features function approximation is interesting. The theoretical intuition about error scaling and control of the error amplification due to Bellman iterates is a good and novel contribution. A weakness of the proposed method is that its effectiveness still depends critically on the availability of good features \phi. The paper could be strengthened if ideas of how interpolation can be applied to other value function approximators were provided. The experimental evaluation is insufficient as only simple problems are considered and not nearly enough experimentation is provided to show empirically the error scaling, the effect of the anchor point selection, the advantage over averagers, or the dependence on the choice of \phi. More results should have been provided on the dependence between the anchor number and location optimization and the choice of features \phi. The theoretical derivation of the error bounds is illuminating in terms of the structure of error scaling but the bounds themselves do not seem practically useful. ## Major Comments + The introduction and preliminaries sections are very well written, giving the right context and a formal problem definition. + The specification of the necessary number of samples to minimize the error between the interpolant and the best linear Q-function approximation in Lemma 2 is nice. The greedy approach to adding anchor points to meet a specified amplification target is a nice contribution. - The greedy heuristic to construct a good set of anchor points before the learning process is a good addition to the proposed algorithm but is not explored in sufficient detail. A discussion on how complex the optimization problem max_{s,a} ||\theta^{\phi(s,a)}||_1 is should be added. Examples of the numbers, positions of anchor points, and the resulting extrapolation coefficient for common choices of \phi would have been nice to see. - The experiments are similar and somewhat simple. It would have been interesting to evaluate LAVIER on a real scale problem in addition to the toy problems vs LS-AVI. For example, it would have been great to see how LAVIER performs on a reinforcement learning problem where a good choice of features is not known but polynomial, RBF, or kitchen sink features are used. A comparison versus averagers and/or OPPQ-Learning [15] would have been interesting as well. It would be have been good to provide more variations on the Chain MDP evaluation, in terms of different feature choices and different planning horizon choices. ## Minor Comments: - Typos: "Whenever the ineherent Bellman error is unbounded," - It would be cleaner to explicitly write the dependence on s_i and a_i in eq. (1) - The meaning of \delta should be introduced more clearly in Lemma 2 - In Fig. 3 middle, what value of C is achieved by the shown anchor points? - The notation is sloppy at places, switching from arguments in parentheses to subscripts for functions or from having two to one subscript (e.g., Proof of Proposition 3, around eq. (2))

[Author Response · NeurIPS 2019]

We thank the reviewers for their comments and insightful reviews.

**=== R1 ===**

**Best approximator in max-norm.** Prior literature (e.g., Remi Munos, *"Error Bounds for Approximate Value Iteration",AAAI-05*, Remi Munos, Csaba Szepesvari, *Finite-Time Bounds for Fitted Value Iteration*, JMLR 2008) studied errors measured in weighted p-norm. This approach is often preferable when using "standard" parametric regression algorithms, as the weighted p-norm can be directly minimized at learning time (e.g., in least-squares regression, the $\ell_2$-norm is minimized). Nonetheless, this gives rise to the so-called concentrability coefficients, which may be very large (even unbounded). In our case, the interpolation algorithm directly controls the max-norm (see e.g., Lemma 1) and this allows us to derive the analysis directly in max-norm and avoid concentrability coefficients.

**=== R2 ===**

**Trade-off amplification and anchor points.** In the worst case even the best trade-off between $\bar{C}$ and $K$ may indeed lead to an exponential complexity. This is not surprising as in the worst case the inherent Bellman error may be unbounded and standard AVI tends to diverge. Recent work (Jinglin Chen, Nan Jiang, *Information-Theoretic Considerations in Batch Reinforcement Learning*, ICML 2019, Conjecture 8) has even conjectured an exponential lower bound in case of unbounded Bellman error. As a consequence, there might exist a fundamental barrier to obtaining polynomial sample complexity in the worst case. Nonetheless, in many other cases a good trade-off between amplification and anchor points may correspond to a much smaller sample complexity (e.g., the condition in Prop.1 could be achieved by a polynomial number of anchor points). Indeed in our experiments even in the case of unbounded Bellman error, the extrapolation can be controlled, and thus we can obtain satisfactory solutions by slightly increasing the number of anchor points without requiring an exponential number of them (see e.g., the experiments on the Tsitsiklis and Van Roy domain and the linear bandit).

**Construction of anchor points.** In the paper we propose a first heuristic algorithm to automatically construct a set of anchor points (see beginning of page 6). While we do not have any guarantee for the method, in our preliminary experiments it seems effective in building a compact set with small amplification factor.

**Incremental construction from $H$ down to 1.** A good choice for the anchor points depends only on the linear architecture, i.e., it is computed exclusively on the basis of the feature map $\phi_t(\cdot, \cdot)$. Thus if one already knows a good set of anchor points at the final timestep $t = H$ and the feature map $\phi_t(\cdot, \cdot)$ does not vary a lot for $t = 1, \ldots, H$ then the set of anchor points for the timestep $t = H$ is also a good choice for prior timesteps $t = 1, \ldots, H - 1$.

**Infinite horizon.** The algorithm does not need any modification (other than the addition of the discount factor) to deal with the infinite horizon case, with the additional benefit that the identification of the support points can be done once at the beginning as opposed to every timestep (as it only depends on the feature map $\phi(\cdot, \cdot)$). However, one would need to change the analysis. Note that Yang and Wang's analysis operates in the much easier setting of zero Bellman error.

**Continuous state space.** The main algorithm can be applied to the continuous state space case without any modification, and the choice of discrete MDPs in the experiments is for illustrative purposes.

**=== R3 ===**

**Computational complexity of the heuristic.** In its most naive form and without further structure, one would need to loop through the state action pairs; for large or continuous state-actions spaces one should sample the state-action pairs to reduce the complexity. This should suffice as we only require approximate convex hulls. For computing the $\theta$'s, this is a linear program and the best known computational complexity can be found in the Arxiv paper "Solving Linear Programs in the Current Matrix Multiplication Time"; other methods, like interior point methods, may be used.

**Experiments**. We can report more thorough statistics for the anchor points (i.e., number and positions, and resulting extrapolation coefficients) as those are computed inside the program. Some of these are reported in figure 3 for that example, but we can add them for the other examples as well.

**Real Life Experiments** As the reviewers points out, we have chosen examples where realizability holds and the choice was deliberate to reduce the number of confounding factors: the approximation error may in general affect the comparison between methods, and might obscure the key underlying processes. We agree with the reviewer that a comparison with averagers like k-NN or kernel based would be practically interesting, but it raises questions of how to best define the function class (for example, the position of the points in a nearest neighbour procedure, the type of radial basis function, etc.) encoded by those settings as a fair comparison to our setting. Therefore in our work we focus on fixing the function approximation class (general linear value functions) and evaluate the impact of the algorithm used to fit the function class. A key benefit of our approach is that it does not modify the underlying (linear) feature representation, allowing the user to use the linear representation with, for example, approximate value iteration, and should this fail, the user can switch to our algorithm and progressively increase the number of support points while keeping the same feature representation. Finally, we can easily add more variations of the chain experiments with varying horizon (for example H = 50, 100, 200) and feature representation as noted by the reviewer.

[Meta-Review · NeurIPS 2019]

All reviewers find the idea of controlling extrapolation error in linear function approximation using interpolation interesting and potentially aspiring for extension to other function approximations. Reviewers do find the empirical evaluation less satisfactory, offering insufficient insight into the strength and/or limitation of the proposed algorithm. Please consider adding supporting experiments requested by reviewers.